# A Multi-Level Analysis of Individual and Neighborhood Factors Associated with Patient Portal Use among Adult Emergency Department Patients with Multimorbidity

**DOI:** 10.3390/ijerph20021231

**Published:** 2023-01-10

**Authors:** Hao Wang, Chan Shen, Michael Barbaro, Amy F. Ho, Mona Pathak, Cita Dunn, Usha Sambamoorthi

**Affiliations:** 1Department of Emergency Medicine, JPS Health Network, Integrative Emergency Services, 1500 S. Main St., Fort Worth, TX 76104, USA; 2Department of Surgery, Penn State Cancer Institute, Hershey, PA 17033, USA; 3Department of Pharmacotherapy, University of North Texas Health Science Center, Fort Worth, TX 76107, USA; 4TCU and UNTHSC School of Medicine, 3500 Camp Bowie Blvd, Fort Worth, TX 76107, USA; 5Texas Center for Health Disparities, Department of Pharmacotherapy, College of Pharmacy, University of North Texas Health Science Center, Fort Worth, TX 76107, USA

**Keywords:** patient portal, multimorbidity, multi-level models, emergency department, adults

## Abstract

Background: Patient portals tethered to electronic health records (EHR) have become vital to patient engagement and better disease management, specifically among adults with multimorbidity. We determined individual and neighborhood factors associated with patient portal use (MyChart) among adult patients with multimorbidity seen in an Emergency Department (ED). Methods: This study adopted a cross-sectional study design and used a linked database of EHR from a single ED site to patients’ neighborhood characteristics (i.e., zip code level) from the American Community Survey. The study population included all adults (age > 18 years), with at least one visit to an ED and multimorbidity between 1 January 2019 to 31 December 2020 (N = 40,544). Patient and neighborhood characteristics were compared among patients with and without MyChart use. Random-intercept multi-level logistic regressions were used to analyze the associations of patient and neighborhood factors with MyChart use. Results: Only 19% (N = 7757) of adults with multimorbidity used the patient portal. In the fully adjusted multi-level model, at the patient level, having a primary care physician (AOR = 5.55, 95% CI 5.07–6.07, *p* < 0.001) and health insurance coverage (AOR = 2.41, 95% CI 2.23–2.61, *p* < 0.001) were associated with MyChart use. At the neighborhood level, 4.73% of the variation in MyChart use was due to differences in neighborhood factors. However, significant heterogeneity existed in patient portal use when neighborhood characteristics were included in the model. Conclusions: Among ED patients with multimorbidity, one in five adults used patient portals. Patient-level factors, such as having primary care physicians and insurance, may promote patient portal use.

## 1. Introduction

Patient portals tethered to patients’ electronic health records (EHR) have become an integral part of the US healthcare system [1,2]. Patient portals allow individuals to manage their healthcare needs through an online interface and facilitate the review of lab results and ordering of medication refills, as well as secure communications with their healthcare providers [3,4]. Studies have reported that patient portal use improved patient-provider communications, increased medical compliance, and yielded higher levels of patient satisfaction, especially among adults with chronic conditions [5,6,7]. The logistical (e.g., tracking their own health information) and psychological (e.g., increased trust in providers) benefits of using a patient portal can increase patients’ healthcare connection, enrich patients’ healthcare engagement experience, and improve patient healthcare satisfaction [8]. In addition, some studies have reported that the use of a patient portal can improve clinical outcomes among patients with certain chronic diseases [9,10].

However, prior literature has shown that the rate of patient portal use is still low (~50%) and varies by patient-level factors, such as age, gender, race and ethnicity, socio-economic status (SES), language, and multimorbidity [11]. Lower rates of patient portal use have been reported among racial and ethnic minorities [12,13]. Sarkar et al. found that Hispanic and Non-Hispanic Blacks (NHB) with diabetes were less likely to use the patient portal compared to Non-Hispanic Whites (NHW) [14]. Lower rates of patient portal use have also been observed among adults with low socioeconomic status (low income and low educational attainment) [15,16]. Adults with higher incomes or higher education levels were more likely to use patient portals than those with low income or less education [17,18]. Localio et al. reported that Spanish-speaking patients with asthma were less likely to use the patient portal than English-speaking patients with asthma [19]. Systematic reviews on facilitators and barriers to patient portal use have been conducted previously [20,21,22]. These reviews suggest that individuals with low income, receiving less education, and without internet access were less likely to use patient portals [23,24].

Few studies have reported variations in the use of patient portals at the neighborhood characteristic level [25,26,27,28]. Gordon et al. reported that patients who lived in communities with higher average income were more likely to use patient portals than those who lived in communities with a lower-than-average income [27]. A study from Balthazar et al. found that patients residing in areas with computers with increased access to internet connections were more likely to use patient portals than those living in areas with less access to the internet [28].

Patients with multimorbidity have been found to be more likely to use patient portals than those without chronic conditions [29,30]. However, variations in patient portal use among adults with multimorbidity are less studied [31]. Understanding the characteristics associated with patient portal use among patients with multimorbidity is important for promoting the use of patient portals and hence strengthening patient self-care management behavior and enhancing the patient–provider relationship, which ultimately may improve patient clinical outcomes [32]. This is particularly important among patients with multimorbidity [32]. The study of adults with multimorbidity seeking care in settings that serve under-represented groups, such as those without health insurance coverage, low socioeconomic status is important, because many may not have access to primary care physicians and only use hospital EDs as their primary medical setting [33]. Therefore, for those who only use EDs for care, ED visits may provide an opportunity to learn about patient portals and trigger the use of patient portals. Additionally, adults with multimorbidity often use EDs because of their needs [34]. For example, in a nationally representative sample of adults with multimorbidity, an overwhelming majority of adults (84.2%) reported using EDs because of the severity of their medical problems [34]. Although not specific to all adults, older adults with composite measures of comorbidity were more likely to use EDs [35]. Under these circumstances, it is important to understand the current patient portal use status among ED patients with multimorbidity. Furthermore, a better understanding of the factors that influence patient portal use at both the individual and the neighborhood level and any potential interactions among adults with multimorbidity is needed. Such information may ultimately help guide innovative implementations and promotion strategies of patient portal use among adults with multimorbidity in the ED. Therefore, the objective of this article is to determine the individual and neighborhood facilitators and barriers associated with patient portal use. 

## 2. Materials and Methods

### 2.1. Study Design and Setting

This is a cross-sectional study. The study site is a single ED of a publicly funded hospital, which is an urban tertiary referral, a level one trauma, and a chest pain and comprehensive stroke center. The study hospital implemented the electronic EHR system (Epic) along with the patient portal (MyChart) in 2012. All patients seeking care in the system have access to their patient portals (MyChart). The ED has an annual volume of approximately 120,000 patient visits. The study procedures comply with the Declaration of Helsinki. This study was approved by the institution’s respective Institutional Review Board with waived informed consent (No. 1600198-5).

### 2.2. Data Source

Data for this study were retrieved by trained data management personnel from the Department of Emergency Medicine and Information Technology. Data validation checks were conducted to ensure accuracy of the data retrieval by randomly selecting 60 patients’ data three times and manually checking these patient records for accuracy. For patients with multiple ED visits, variables were extracted from the last recorded visit during the study period.

### 2.3. Study Population

The study population comprised patients who had at least one ED visit between 1 January 2019 to 31 December 2020. During this period, 121,044 patients made 238,723 visits. The number of visits per person ranged from 1 to 231, with a median of 1 visit (interquartile range of 1–2).

### 2.4. Inclusion and Exclusion Criteria

For this study, we included adults (age ≥ 18 years) with multimorbidity. Multimorbidity was defined as the presence of two or more chronic conditions based upon the Goodman’s criteria [36].

We excluded adults with missing information on any of the following variables: (1) sex (2); marital status (3); preferred language (4); insurance status; and (5) patient zip code. We further excluded patients who had not visited the study hospital (i.e., any ED, clinical visit, or hospitalization) 12 months prior to their indexed ED visits, because these patients may not have access to the patient portal. 

We also excluded zip codes with less than 50 patients, because we adopted a multi-level model analysis (i.e., patients nested within zip codes (i.e., neighborhoods)) [37]. As we were interested in the interactive associations of race, ethnicity, and language with patient portal use (i.e., patient portal has only a English or Spanish version), we excluded approximately 1% of the patients (*n* = 528) for the following reasons: these patients were either NHW who did not speak English (*n* = 146); NHB who did not speak English (*n* = 304); other races who spoke Spanish (*n* = 48); or Hispanics who spoke other languages (i.e., neither English nor Spanish, *n* = 30). The final study population consisted of 40,544 adult patients with multimorbidity. Details of population selection at each step are shown in Figure 1. 

### 2.5. Measures

Dependent Variable: Patient Portal Use (Yes/No): 

We created a binary indicator for patient portal use by identifying patients who had a MyChart account and used MyChart service at least once 12 months before the index ED visit. Core functions of patient portal was shown in Appendix B Figure A1.

Explanatory Variables: 

Patient-level variables: These included age, sex, marital status, race and ethnicity, preferred language, health insurance, and visits to a primary care physician. We divided age into five groups based on the age group distribution recommended by HINTS (Health Information National Trend Survey) [18] (i.e., 18–34 years, 35–49 years, 50–64 years, 65–74 years, and 75 or older); sex into two groups (male and female); marital status into four groups (single, married, divorced, and others); two groups of health insurance status (yes or no); and whether patients had a primary care physicians (yes or no). We also combined race and ethnicity information (NHW, NHB, Hispanic/Latino, and other race) with preferred language (English, Spanish, other).

Neighborhood-level variables: These were derived from the 2020 American Community Survey (ACS) that included social (table name: DP02 selected social characteristics in the United States); economic (table name: DP03 selected economic characteristics); housing (table name: DP04 selected housing characteristics); and demographics (table name: DP05 ACS demographics and housing estimates) characteristics at the zip code level. We linked the ACS 5-year estimates at the zip code level to the patients’ zip code recorded in the EHR. The number of residents by zip code ranged from 50 to 3191.

We extracted the following neighborhood characteristics: (1) percentage unemployed; (2) median family income; (3) percentage of family below poverty levels; (4) percentage of Hispanic residents; (5) percentage of residents reporting two or more races; (6) percentage of people speaking other languages; (7) percentage of residents receiving Bachelor or higher degrees; (8) percentage of residents without health insurance coverage; (9) percentage of married couples; (10) percentage of households with internet access; (11) percentage of non-US citizens; and (12) percentage of households with no vehicle in the family. We selected these neighborhood characteristics based on published studies suggest their associations with digital health use [38,39].

As there were many neighborhood factors representing each of the domains, we analyzed the correlations among the neighborhood-level explanatory variables (Appendix A). We considered the Pearson correlation co-efficient estimates with r ≥ 0.5 indicating a relatively high correlation, 0.3 < r < 0.5 indicating a mild-to-moderate correlation, and r ≤ 0.3 indicating a weak correlation. We found that poverty status had a high correlation with nearly all other variables. Therefore, poverty was chosen as one of the neighborhood characteristics. In addition, we chose two other neighborhood characteristics (i.e., percentage of people speaking other languages and percentage of residents without health insurance coverage) to match patient-level variables (i.e., insurance status and preferred language).

### 2.6. Statistical Analyses

Unadjusted and adjusted associations of patient-level and neighborhood-level factors with the patient portal use were analyzed, using multi-level logistic regressions with random intercepts. We used the multi-level approach because the patients were clustered within zip codes. In multi-level models, the residual variance is partitioned into “between neighborhood” (i.e., zip code level) and a “within neighborhood” (i.e., patient-level) variance. The model estimates regression coefficients of the explanatory variables, known as the fixed effects. 

We performed three nested models: Model 1 (Null model) estimated the “between neighborhood” variation in patient portal use, which incorporated only “neighborhood-specific” random effects; Model 2 adjusted for all patient-level characteristics mentioned in the measures section with a random intercept; Model 3 added neighborhood characteristics or contextual factors (percentage of residents below federal poverty line, percentage of residents speaking other languages, and percentage of residents without health insurance coverage).

To investigate the extent to which individual-level patient portal use was statistically dependent on the area of residence, we used variance partition coefficient (VPC). Higher values of VPC indicate greater variation among neighborhoods. As the VPC is dependent on the prevalence of outcome, we used Median Odds Ratio (MOR) to better understand the extent to which the individual’s patient portal use is determined by neighborhood. A MOR of 1 represents no differences in patient-portal use between neighborhoods and values greater than 1 suggests that strong neighborhood differences [40].

In a random effects logistic regression model, interpreting the association of neighborhood variables with patient portal use needs to consider the neighborhood-level variance (i.e., random effects). Therefore, we used the 80% interval odds ratio (IOR-80), and proportion of opposed odds ratios (POOR) [41,42]. We used IOR-80 to explain the regression coefficients of the neighborhood variables along with the residual variance. A wide IOR-80 of a neighborhood variable that includes one indicates that the specific neighborhood variable is not important in understanding the neighborhood-level variations in the patient portal use. POOR is the proportion of odds ratios in the opposite direction to the overall odds ratios of the neighborhood variables [43]. A POOR of 50% indicated that half of the pair-wise comparisons of odds ratios (associations of specific neighborhood characteristic with patient portal use) are in the direction opposite to the overall odds ratio [43].

In this study, we used STATA 14.2 (College Station, TX, USA) for data analysis. 

### 2.7. Reporting Guideline

We followed Strengthening of The Reporting of Observational Studies in Epidemiology (STROBE) reporting guidelines in describing our study methods and findings [44].

## 3. Results

The study population had an even distribution of males and females (50%) (Table 1). Approximately one-third (35.5%) of adults with multimorbidity were NHW. The most preferred language was English, with only 11.3% indicating preference for either Spanish or other languages (2.1%). An overwhelming majority of patients (82%) were non-elderly (less than 65 years). 

Table 2 summarizes neighborhood characteristics based on the 75 zip codes. We observed substantial variations in neighborhood characteristics. For example, the percentage of Hispanic residents ranged from 4.0% to 93.9% (Median = 36.3%); residents speaking languages other than English ranged from 6.3% to 81.6%. Similarly, the percentage of those with no health insurance ranged from 3.0% to as high as 41.0%. Most of the zip codes had internet access, with a median value of 81.2 and mean of 79.1%. The average percentage below the poverty level among all zip codes was 18.3%. 

Overall, 19.1% of patients accessed patient portals (Table 3). Patient portal use was also positively related to the number of patient ED visits. We found that 15.74% (2663/16,919) of patient portal users were patients who only visited the ED once during the study period; 17.85% (1666/9335) of patient portal users were patients who visited the ED twice; 22.63% (1122/4959) of patient portal users were patients visited the ED thrice; and 24.71% (2306/9331) of patient portal users were patients who visited the ED more than three times (*p* < 0.0001). A significantly lower percentage of older patients (aged 65 years or older), males, NHB, Hispanics, and individuals whose preferred language is Spanish used patient portals compared to young adults, females, NHW, and whose preferred language is English. However, a higher percentage of individuals with health insurance or had a primary care physician visit used patient portals (Table 3). At the neighborhood level, we observed that less patient portal users lived in communities with a higher percentage of Hispanics, a higher percentage of residents who speak other languages, higher poverty rates, and lower rates of college education and internet access.

Measures of variations (random effects) in patient portal use: 

We observed that 7.1% of the variation in patient portal use was due to differences between neighborhoods (Table 4). In Model 2, after adjusting for patient-level variables, the variation dropped to 6.0%. In Model 3, the percentage variation was reduced to 4.1% after including the three neighborhood characteristics. The MOR from Models 2 and 3 were greater than 1 (Model 2 MOR = 1.55, 95% CI = 1.43, 1.68, Model 3 MOR = 1.43, 95% CI = 1.33, 1.53) suggesting that the neighborhood variations are important in determining the variations in patient portal use of the individuals. 

Associations of individual-level measures (fixed effects) with the patient portal use: 

All patient-level variables were significantly associated with patient portal use. However, these associations are conditional on both other explanatory variables in the model and the neighborhood-specific random effect. For example, within a neighborhood, the odds of patient portal use for an individual with a PCP visit was 5 times the odds of an individual without a PCP visit (AOR = 5.54, 95% CI 5.06–6.06, *p* < 0.001), but share identical values on the remaining explanatory variables and has the same value of the random effect (i.e., neighborhood average risk). Within a community, older adults (aged 75 years or older) were less likely to use (AOR = 0.53, 95% CI = 0.45, 0.61) patient portals compared to younger adults. Within a community, Spanish-speaking Hispanic adults and English-speaking NHB were less likely to use patient portals compared to the NHW (Table 4). 

Association of neighborhood variables with the patient portal use: 

Because the values of neighborhood variables are constant for all individuals living in the same neighborhood, the interpretation of odds ratios associated with the neighborhood characteristics is challenging. Therefore, we rely on IOR-80 and POOR estimates for interpretation. The 80% IORs for the three neighborhood characteristics in Model 3 were (0.50, 1.97) for a unit increase in percentage below the federal poverty level; (0.48, 1.86) for a unit increase in the percentage with no health insurance; and (0.52, 2.02) for a unit increase in the percentage speaking other languages. For all three neighborhood characteristics, the IOR-80 estimates contain one, suggesting that the unexplained neighborhood variations are stronger than the specific neighborhood poverty, insurance, and spoken language effects (Table 4). The widths of the intervals also suggest that adding the neighborhood variables did not meaningfully explain variations in patient portal use.

In our study, the POOR estimates for neighborhood poverty, health insurance coverage, and speaking other languages were: 0.50, 0.45, and 0.48, respectively (Table 4). For example, in 45% of comparisons for one unit increase in uninsured rates, the odds ratios would be in a different direction to that of the overall odds ratio for no insurance rates. These findings suggest that heterogeneity in the association of neighborhood variables and patient portal use was very high. The overall odds ratio for neighborhood uninsurance variable was 0.94, denoting lower odds of patient portal use in neighborhoods with high rates of uninsurance.

## 4. Discussion

In this study, one in five adults (19.1%) used the patient portal over a 12-month period, suggesting that patient portal uptake is very low among adults with multimorbidity. The rate of patient portal use was low in our study in comparison to other reports among patients with multimorbidity or certain common chronic conditions, such as hypertension and diabetes [45,46,47]. This could be partially explained by the vulnerable patient population, due to the nature of the study’s hospital setting (i.e., publicly funded urban hospital with patients of low SES and higher poverty levels when compared with national averages) [48]. A similar rate of patient portal use has been documented among individuals with SES and high poverty [49]. Therefore, we suggest advocating patient portal use from ED physicians might need to be emphasized, especially when targeted at such patient population. 

At the patient level, we observed Spanish-speaking Hispanic/Latino patients and NHB were less likely to use patient portals compare to NHW. Racial and ethnic disparities have been documented in prior studies of HIT use [14,50]. Specifically, NHB and Hispanic/Latino patients were less likely to use patient portals [12,13,14]. The explanations for the racial and ethnic disparity in HIT use include low socioeconomic status (SES), such as having high rate of poverty, receiving less educational training, or having limited internet access [12,13,14]. In our study, we adjusted for neighborhood SES (poverty and health insurance coverage) and we observed substantial variations in the relationship between neighborhood variables and patient portal use. Although we did not explore communication challenges, their role in patient portal use cannot be ruled out. For example, other studies have reported that NHB and Hispanic/Latino patients were less likely to be enrolled in patient portal by healthcare providers due to communication issues with their healthcare providers [15,51].

There were no significant differences in patient portal use between Hispanic/Latino patients who speak English and NHW patients. In our study, Hispanic/Latino patients who speak English were younger and more likely to have insurance coverage than those who speak Spanish (Appendix A). This may, in part, explain the high rate of patient portal use among Hispanic/Latino English-speaking patients compared to their Spanish-speaking counterparts. Our findings call for further research exploring the barriers to patient portal use by Spanish-speaking Hispanic/Latino patients [19]. This is because the patient portal is available in both English and Spanish in the EHR system. 

Patient-level health insurance coverage and PCP visits were positively associated with the patient portal use. PCP visits may represent having a regular healthcare provider. A study using data from insured respondents derived from the 2017 Health Information National Trends Survey reported that those with usual care were more likely to use patient portals [18,47,49]. Studies also have reported that PCPs and their team members encourage patients to use patient portals [52,53]. As patients with health insurance are more likely to have a usual source of care or a PCP, it is not surprising that the presence of health insurance was associated with patient portal use. 

Our study findings highlight the importance of neighborhood or contextual effects on patient portal use. The AORs of patient-level variables were outside the MOR range, suggesting that patient-level variables may be more important than variations in patient portal use across neighborhoods. The IOR-80 suggested that the neighborhood-level variations were too strong and individual neighborhood factors were not able to distinguish between presence and absence of patient portal use. Adding the neighborhood variables did not explain variations in neighborhood-level patient portal use. It is possible that other neighborhood-level factors, such as social capital or social cohesion, may affect health information technology use, including patient portal use, which were not available to the study investigators. For example, social capital may affect patient portal use through different pathways: (1) by providing access to patient portal use through robust internet coverage in the neighborhood [27,54]; (2) by expanding access to health information technology by installing health kiosks in targeted locations (example: local departments of social service, and community health centers) [55]; (3) dissemination of resources (for example: communities that are cohesive, in which neighbors trust each other, may more greatly and easily diffuse the benefits of using patient portals); and (4) by providing support (neighbors may take it as a civic responsibility and provide support services for patient portal use). Therefore, an important agenda for future research is to identify the associations of neighborhood factors such as social capital, which are not readily available in secondary data sources, with health information technology in general and patient portal use specifically. 

This study has several unique strengths. We used electronic health records to explore digital health technology use among vulnerable patient populations, such as minorities with multimorbidity. By focusing on ED patients, the study focuses on the ED as a venue for promoting patient portal use. Meanwhile, we also examined the contribution of both the individual- and neighborhood-level variables. Inevitably, our study has limitations. First, this is a single-center retrospective observational study. Therefore, our findings required external validations. Second, we could not include all potential variables that could potentially affect patient portal use at both the individual and the neighborhood levels. At the neighborhood characteristic analysis, we were not able to determine the association of neighborhood solidarity or social capital with patient portal use, due to limited data resource. We chose variables to include in the current analysis based upon previous studies’ findings and experts’ opinions [28,29], which may generate potential biases. In addition, our study results were based on the patients who visited the study’s healthcare system; therefore, we were unable to ascertain the patient portal status among patients who visited multiple hospital systems. Third, we determined patients who used the patient portal at least once as positive patient portal users. However, differences certainly occurred between those who only used the patient portal once versus those who used it many times during the study period. In this study, we were unable to determine such differences. Additionally, we determined patient portal use based on the past 12-month history and paired it with individual patients’ demographic and clinical information during their indexed ED visits. Using such a method could potentially generate incorrect information if patient demographic or clinical information changed significantly in the past 12 months before their index ED visit (e.g., we pair a patient’s zip code with community variables; however, if a certain patient moved during the past 12 months, their patient portal use should be paired with the previous zip code for analysis). However, we randomly checked 50 patients who had multiple ED encounters and found a relatively low rate of patients changing zip codes (2%). Lastly, this study did not analyze other variables that might also be associated with the patient portal use, such as patient zip code distance to hospital, patients’ access to transportation, or patients’ access to their primary care physicians, etc. Therefore, our findings warrant future larger-scale prospective studies for external validation.

## 5. Conclusions

Patient portal use was low among emergency department patients with multimorbidity. Disparities in digital technology use were apparent, especially among Non-Hispanic Black and Hispanic/Latino Spanish-speaking patients. Primary care physician visits might be an opportunity to empower patients with digital technology use. 

## Figures and Tables

**Figure 1 ijerph-20-01231-f001:**
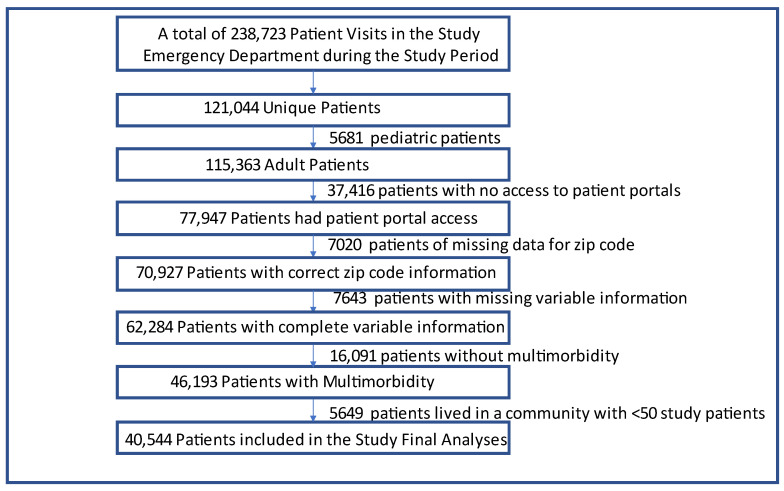
Study flow diagram.

**Table 1 ijerph-20-01231-t001:** Patient-level characteristics of adults with multimorbidity (N = 40,544), linked Electronic health records and American Community Survey Database, 2019–2020.

		Number	Percentage
Sex			
	Female	20,234	49.9
	Male	20,310	50.1
Race, Ethnicity, and Preferred Language			
	Non-Hispanic White and English	14,385	35.5
	Non-Hispanic Black and English	13,609	33.6
	Latino/Hispanic and English	6011	14.8
	Latino/Hispanic and Spanish	4603	11.4
	Other race and English	1084	2.7
	Other race and Other language	852	2.1
Age Groups			
	18–34 years	6619	16.3
	35–49 years	10,183	25.1
	50–64 years	16,416	40.5
	65–74 years	5046	12.5
	75 years+	2280	5.6
Marital Status			
	Single	19,163	47.3
	Married	10,185	25.1
	Divorced	5432	13.4
	Other	5764	14.2
PCP Visit			
	Yes	28,046	69.2
	No	12,498	30.8
Health Insurance			
	Yes	30,165	74.4
	No	10,379	25.6

Note: Based on 40,544 adults aged 18 years or older with multimorbidity, at least one visit to the emergency department in 2019–2020, and access to a patient portal in the previous 12 months. PCP: primary care physician.

**Table 2 ijerph-20-01231-t002:** Community-level social (DP02), economic (DP03), and demographics (DP05) characteristics percentages at the zip code-level, American Community Survey 2019–2020.

	Mean	STD	Median	Min	Max	IQR
Unemployment	5.6	1.4	5.5	1.5	9.2	[4.7–6.9]
Median family income in US $	63,390	22,304	60,284	36,300	177,393	[43,042–80,040]
Below poverty level	18.3	8.7	18.7	2.7	33.9	[10.6–25.8]
Hispanic residents	38.5	19.7	36.3	4.0	93.9	[23.7–47.8]
Residents reporting two or more races	2.7	1.0	2.6	0.6	5.4	[2.0–3.6]
People speaking other languages	36.3	18.0	30	6.3	81.6	[21.4–46.2]
Bachelor or higher degrees	22.6	13.3	20.6	4.2	72.3	[12.0–31.9]
No health insurance	21.9	8.3	22.3	3.0	41.0	[14.9–28.1]
Married couples	43.7	11.4	42.2	24.8	77.8	[34.3–51.6]
Access to Internet	79.1	11.1	81.2	56.8	98.8	[68.8–87.1]
Non-US citizen	66.0	14.4	71.6	24.3	81.5	[54.0–77.0]
No vehicle in family	6.6	3.7	6.8	0.4	14.4	[3.4–9.3]

Note: Based on resident zip codes of 40,544 adults aged 18 years or older with multimorbidity, at least one visit to the emergency department in 2019–2020, and access to a patient portal in the previous 12 months.

**Table 3 ijerph-20-01231-t003:** Number and row percentages of adults with multimorbidity with ED visits by patient portal (My Chart) use, linked electronic health records with American Community Survey database, 2019–2020 (*N* = 40,544).

		Patient Portal Users		Non-Users			
		*N*	%	*N*	%	Chi^2^ Value	*p*-Value
Patient-Level Characteristics							
ALL		7757	19.1	32,787	80.9		
Sex						712.1	<0.001
	Female	4928	24.4	15,306	75.6		
	Male	2829	13.9	17,481	86.1		
Race, Ethnicity, and Preferred Language						468.9	<0.001
	Non-Hispanic White and English	3191	22.2	11,194	77.8		
	Non-Hispanic African American and English	2121	15.6	11,488	84.4		
	Latino/Hispanic and English	1258	20.9	4753	79.1		
	Latino/Hispanic and Spanish	605	13.1	3998	86.9		
	Other race and English	311	28.7	773	71.3		
	Other race and Other language	271	31.8	581	68.2		
Age Groups						183.9	<0.001
	18–34 years	971	14.7	5648	85.3		
	35–49 years	2085	20.5	8098	79.5		
	50–64 years	3462	21.1	12,954	78.9		
	65–74 years	928	18.4	4118	81.6		
	75 years or older	311	13.6	1969	86.4		
Marital Status						326.1	<0.001
	Single	2975	15.5	16,188	84.5		
	Married	2346	23.0	7839	77.0		
	Divorced	1262	23.2	4170	76.8		
	Other	1174	20.4	4590	79.6		
PCP						>2400	<0.001
	Yes	7158	25.5	20,888	74.5		
	No	599	4.8	11,899	95.2		
Health Insurance						977.6	<0.001
	Yes	6852	22.7	23,313	77.3		
	No	905	8.7	9474	91.3		
**Community-Level Characteristics**							
		**Mean**	**SE**	**Mean**	**SE**	**t-Value**	***p*-Value**
	Married couple (%)	45.90	0.13	43.16	0.06	19.10	<0.001
	None US citizen (%)	62.39	0.17	66.87	0.08	24.89	<0.001
	Unemployment (%)	5.36	0.02	5.67	0.01	18.63	<0.001
	Median family income ($)	67,468.99	260.19	62,425.30	121.76	17.98	<0.001
	No insurance (%)	20.22	0.09	22.24	0.05	19.43	<0.001
	Below poverty level (%)	16.15	0.09	18.84	0.05	24.72	<0.001
	No vehicle in family (%)	5.64	0.04	6.77	0.02	24.60	<0.001
	Multiracial communities (%)	2.89	0.01	2.67	0.01	17.75	<0.001
	Hispanic residents (%)	35.66	0.22	39.18	0.11	14.20	<0.001
	People speaking other languages (%)	34.26	0.20	36.80	0.10	11.17	<0.001
	Bachelor or higher degrees (%)	24.39	0.15	22.18	0.07	13.17	<0.001
	Access to Internet (%)	81.61	0.12	78.44	0.06	22.82	<0.001

Note: Based on 40,544 adults aged 18 years or older with multimorbidity, at least one visit to the emergency department in 2019–2020, and access to a patient portal in the previous 12 months. Patient-level characteristics were derived from the electronic health records and community-level characteristics were derived from the American Community Survey. Group differences in categorical variables by patient portal use were tested with chi-square tests and group differences in continuous variables by patient portal use were tested with *t*-tests.

**Table 4 ijerph-20-01231-t004:** Random intercept multi-level logistic regressions on patient portal use adults with multimorbidity, linked electronic health records with zip code-level American Community Survey, 2019–2020.

Random Effects	Estimate of Variance of the Random Intercept	Variation Partition Coefficient (%)
Model 1: Null model	0.25	7.16
Model 2: Adjusted for patient-level factors	0.21	6.10
Model 3: Adjusted for both patient-level and neighborhood-level factors	0.14	4.12
	Model 2 AOR (95%CI)	Model 3 AOR (95% CI)
Sex (Reference Group—Male) Female	1.77 [1.68–1.87]	1.77 [1.68–1.87]
Race/ethnicity and preferred language (Reference group—NHW speaking English) Non-Hispanic Black and English Hispanic/Latino and English Hispanic/Latino and Spanish Other races and English Other races and other languages	0.66 [0.61–0.70] 0.94 [0.86–1.02] 0.50 [0.45–0.56] 1.02 [0.87–1.18] 0.88 [0.74–1.04]	0.66 [0.61–0.71] 0.94 [0.86–1.02] 0.50 [0.45–0.56] 1.01 [0.87–1.18] 0.87 [0.73–1.02]
Age Groups (Reference Group = 18–34 years) 35–49 years 50–64 years 65–74 years 75 years +	1.15 [1.05–1.27] 0.95 [0.87–1.04] 0.74 [0.66–0.83] 0.53 [0.45–0.61]	1.16 [1.05–1.27] 0.9 [0.87–1.04] 0.74 [0.66–0.83] 0.53 [0.45–0.61]
Marital Status (Reference Group = Single) Married Divorced Others	1.47 [1.37–1.57] 1.29 [1.19–1.40] 1.23 [1.13–1.33]	1.47 [1.37–1.57] 1.29 [1.19–1.40] 1.22 [1.13–1.33]
PCP Visit (Reference Group = No) Yes	5.55 [5.07–6.07]	5.54 [5.06–6.06]
Health Insurance (Reference Group = No) Yes	2.41 [2.23–2.61]	2.41 [2.22–2.61]
Neighborhood Factors		
Measure of neighborhood variation MOR	1.55 [1.43–1.68]	1.43 [1.33–1.53]
Neighborhood factors (Associations) Percent residents below federal poverty level AOR (95%, *p*) IOR-80% POOR (%) Percent with no health insurance AOR (95%, *p*) IOR-80% POOR (%) Percent speaking other languages AOR (95%, *p*) IOR-80% POOR (%)		0.99 [0.97–1.02] 1.00 [0.50–1.97] 50 0.94 [0.91–0.97] 0.94 [0.48–1.86] 45 1.02 [1.01–1.04] 1.02 [0.52–2.02] 48

Note: Based on 40,544 adults aged 18 years or older with multimorbidity, at least one visit to the emergency department in 2019–2020, and access to a patient portal in the previous 12 months. Patient characteristics were derived from the electronic health records and neighborhood characteristics were derived from the American Community Survey. Abbreviations: AOR: adjusted odds ratio; CI: confidence interval; IOR: interval odds ratios; MOR: median odds ratio; PCP: primary care physician; POOR: proportion of opposed odds ratio; VPC: variance partition co-efficient.

## Data Availability

Data available upon request to the first and corresponding authors.

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
