# Peer review of "A Multi-Level Analysis of Individual and Neighborhood Factors Associated with Patient Portal Use among Adult Emergency Department Patients with Multimorbidity"

_ijerph, 2023, doi:10.3390/ijerph20021231_

Round 1

Reviewer 1 Report

Dear authors

This is a well written paper with sound methodology and an interesting reserach question.

I have some minor remarks which might imporve the paper after adjustments.

Introduction

It can be better justified why ED visits are important for patients with multi morbidity. Now it comes across a bit too much as a convenience sample to  take these ER visits. 

Methodology

You write in the last sentence of setting that also patients with multiple vistits are included. This triggers the question (for me) if patients with higher amount of ER vistits are the ones that use the patient portal? 

Visits per patients could be as high as 231? (this means that there was a visit once/ week?) maybe it would be better to state the median and the min and max between (). 

In the inclusion and exclusion criteria a part of the sentence is missing I think? we excluded... any of the following variables: 1) ?

Exclusion of zip codes less than 50 patients/ group. I would like to see some references supporting this choice, or some statistical power analysis. 

Other languages excluded... maybe I missed it but I had to wait till the discussion part that the portal was in Spanish and English.

measures: in general I think 1 visit to the prortal is actually rather low to be called an active user, this might not lead to the listed advantages. I would discuss that in the discussion. 

neighbourhood-level variables: what are those table names? (eg table name: DP02)

Discussion

see previous remarks
+ what are the practical implications for the findings? 

PCP involvement: does the GP has acces to the hospitals portal, more specific the patients portal? what if the patients visits several hospitals? All barriers to include the PCP.

Good luck with the paper!

Author Response

We thank the reviewer for their valuable feedback. The reviewer’s comments are italicized and our responses to the reviewer’s comments are in blue font. We are also sharing a revised manuscript with track changes of all the updates.

Review #1

This is a well written paper with sound methodology and an interesting research question.

Thank you for the positive feedback.

I have some minor remarks which might improve the paper after adjustments.

Introduction

It can be better justified why ED visits are important for patients with multi morbidity. Now it comes across a bit too much as a convenience sample to take these ER visits. 

In response to the reviewer’s comment, we have provided a robust justification of why ED visits are important for patients with multimorbidity.  The revised text is given below.

In our study, we restricted the study population to ED adult patients with multimorbidity for the following reasons:

First, the study site is a publicly funded hospital and serves under-represented groups such as those without health insurance coverage, low-socio economic status; many of them may not have access to primary care physician and patients only use hospital ED as their primary medical home.  For example, nearly 30% of the study population had only ED visits. Second, for those who only use ED for care, ED visits may provide an opportunity to learn about patient portals and trigger use of patient portals.  Third, adults with multimorbidity often use EDs because of their needs.  For example, in a nationally representative sample of adults with multimorbidity an overwhelming majority of adults (84.2%) reported using ED because of seriousness of medical problems (Psychiatry J. 2017; 2017:8565186). Although not specific to all adults, older adults with composite measures of comorbidity were more likely to use ED ( McCusker J, Karp I, Cardin S, Durand P, Morin J. Determinants of emergency department visits by older adults: a systematic review. Acad Emerg Med. 2003;10(12):1362–70). Fourth, there is published literature using data from outpatient clinics. We realized that we did not address enough in our introduction, so we revised our introduction section with the addition of such rationale.  (see line 78-90)

Methodology

You write in the last sentence of setting that also patients with multiple visits are included. This triggers the question (for me) if patients with higher amount of ER visits are the ones that use the patient portal? 

Response:

We do acknowledge that greater number of ER visits may provide more opportunities for exposure and training of patient portal use. Due to the benefits of patient portal use, we believe that even one visit may provide an opportunity to learn about patient portals.

To address the reviewer’s comment, we added the % with patient portal use by number of ED visits (see Line 228-234).

Number of patients

Percentage of patient portal use

Patients with only 1 ED visit

16,919

15.74% (2663)

Patients with 2 ED visits

9,335

17.85% (1666)

Patients with 3 ED visits

4,959

22.63% (1122)

Patients with more than 3 ED visits

9,331

24.71% (2306)

Visits per patients could be as high as 231? (this means that there was a visit once/ week?) maybe it would be better to state the median and the min and max between (). 

Response: Yes, there was one patient who was the highest ED user and who visited ED 231 times during the study period. We did not exclude this patient to reflect the real-world situation.  We now include median and IQR of number of ED visits (see Line 117-118).

In the inclusion and exclusion criteria a part of the sentence is missing I think? we excluded... any of the following variables: 1) ?

Response:  We apologize for the omission.  The manuscript missed one full sentence.  We have now revised the sentence as follows:

We excluded adults with missing information on any of the following variables: 1) sex, 2) marital status, 3) preferred language, 4) insurance status, and 5) patient zip code. (see line 124).

Exclusion of zip codes less than 50 patients/ group. I would like to see some references supporting this choice, or some statistical power analysis. 

Response: We revised our manuscript with the addition of the following reference. (see line 129)

Maas CJM and Hox JJ. Sufficient sample sizes for multilevel modeling. Methodology, 2005;1(3): 86-92.

Other languages excluded... maybe I missed it but I had to wait till the discussion part that the portal was in Spanish and English.

Response: We now mention that the patient portal only has English or Spanish version in the methods section (see line 131).

measures: in general I think 1 visit to the prortal is actually rather low to be called an active user, this might not lead to the listed advantages. I would discuss that in the discussion. 

Response: Thanks for the reviewer’s valued comment. We realized such limitations. We revised and addressed it in the limitation section. (see line 352-358)

neighbourhood-level variables: what are those table names? (eg table name: DP02)

Response: Yes, we added the table names. (see line 154-157)

Discussion

see previous remarks
+ what are the practical implications for the findings? 

Response: Yes, we revised and addressed the practical implications in the findings (see line 291-292).

PCP involvement: does the GP has acces to the hospitals portal, more specific the patients portal? what if the patients visits several hospitals? All barriers to include the PCP.

Response: Yes, the GP has access to the hospital portal and patients have access to the patient portal. Patients may visit several hospitals and this study only determines the patient portal use at the study healthcare system, which is addressed in the limitation section.

As mentioned previously, the system serves under-represented population with low socioeconomic status, may not have access to PCPs and may use ED as their primary medical home.  Please see  line 352-358 and line 78-90).

Reviewer 2 Report

A brief summary

Electronic health records are the main way in which digitisation is being applied to health services. This study focuses on the factors influencing the use of MyChart by patients with multiple chronic conditions and compares the characteristics of patients in this neighbourhood using single acute care data as a database. The study is innovative and provides a reference for future measures to implement digital health services in the community.

Point 1

Line 56-57

Health services are divided into two main types of services, outpatient and emergency, the patients with multiple chronic conditions generally using outpatient clinics more often. The authors observed that socio-economic factors were the main reason affecting patients' use of MyChart in outpatient clinics, but did not to explain the reasons for including emergency patients as the study population in this study. I would suggest that the authors add an explanation as to why emergency patients were used as the study subjects.

Point 2

Line 127-128

Having a MyChart account is not the same as using the MyChart service, I presume the author means having a MyChart account and using it at least once? Could the author please confirm if we understand correctly?

Point 3

Line 206-207

This study describes the characteristics of 75 zip codes, but does not describe the distance of these communities from the publicly funded hospital. This relates to healthcare accessibility. Would patients choose to access healthcare directly rather than through MyChart if they were closer? I would suggest that the authors explain the reasons why medical accessibility would not have an impact on the results of this study.

Point 4

The title of this study is a multilevel analysis of personal and neighbourhood relationships, and the reader would have preferred to see whether elements such as neighbourhood solidarity or social capital had an impact on patients' use of MyChart. However, the authors describe relatively little of the neighbourhood effect in their discussion. I would suggest that the authors add to the discussion other elements of the neighbourhood that may have had a positive or negative impact on the use of MyChart.

Author Response

We thank the reviewer for their valuable feedback. The reviewer’s comments are italicized and our responses to the reviewer’s comments are in blue font. We are also sharing a revised manuscript with track changes of all the updates.

Review #2 comment

A brief summary

Electronic health records are the main way in which digitisation is being applied to health services. This study focuses on the factors influencing the use of MyChart by patients with multiple chronic conditions and compares the characteristics of patients in this neighbourhood using single acute care data as a database. The study is innovative and provides a reference for future measures to implement digital health services in the community.

Point 1

Line 56-57

Health services are divided into two main types of services, outpatient and emergency, the patients with multiple chronic conditions generally using outpatient clinics more often. The authors observed that socio-economic factors were the main reason affecting patients' use of MyChart in outpatient clinics, but did not to explain the reasons for including emergency patients as the study population in this study. I would suggest that the authors add an explanation as to why emergency patients were used as the study subjects.

Response: We have added why emergency patients were used as the study subjects.

In our study, we restricted the study population to ED adult patients with multimorbidity for the following reasons:

First, the study site is a publicly funded hospital and serves under-represented groups such as those without health insurance coverage, low-socio economic status; many of them may not have access to primary care physician patients only use hospital ED as their primary medical home.  For example, nearly 30% of the study population had only ED visits. Second, for those who only use ED for care, ED visits may provide an opportunity to learn about patient portals and trigger use of patient portals.  Third, adults with multimorbidity often use EDs because of their needs.  For example, in a nationally representative sample of adults with multimorbidity an overwhelming majority of adults (84.2%) reported using ED because of seriousness of medical problems (https://www.ncbi.nlm.nih.gov/pmc/articles/PMC5612322/). Although not specific to all adults, older adults with composite measures of comorbidity were more likely to use ED ( McCusker J, Karp I, Cardin S, Durand P, Morin J. Determinants of emergency department visits by older adults: a systematic review. Acad Emerg Med. 2003;10(12):1362–70). Fourth, there is published literature using data from outpatient clinics We realized that we did not address enough in our introduction, so we revised our introduction section with the addition of such rationale.  (see line 78-90)

Point 2

Line 127-128

Having a MyChart account is not the same as using the MyChart service, I presume the author means having a MyChart account and using it at least once? Could the author please confirm if we understand correctly?

Response: we determined positive patient portal use among patients who had a MyChart account and used MyChart account at least once. We revised it (see line 140-141).

Point 3

Line 206-207

This study describes the characteristics of 75 zip codes, but does not describe the distance of these communities from the publicly funded hospital. This relates to healthcare accessibility. Would patients choose to access healthcare directly rather than through MyChart if they were closer? I would suggest that the authors explain the reasons why medical accessibility would not have an impact on the results of this study.

Response:  We acknowledge that distance to ED facilities may have an impact on healthcare use.  There is some evidence that older adults living within 10km of an ED facility were more likely to be frequent ED users (Franchi C, Cartabia M, Santalucia P, Baviera M, Mannucci PM, Fortino I, et al. Emergency department visits in older people: pattern of use, contributing factors, geographical differences and outcomes. Aging Clin Exp Res. 2017. https://doi.org/10.1007/s40520-016-0550-5.)

However, the influence of location on ED use is complicated because ED users are not a homogenous group. Patients admitted ED may have been brought by ambulance, may have private vehicles, or may have used public transportation.  As our focus is on patient portal use once admitted to the ED, there may not be a direct association of location to patient portal use. We addressed such in our limitation as well (see line 366-369).

Point 4

The title of this study is a multilevel analysis of personal and neighbourhood relationships, and the reader would have preferred to see whether elements such as neighbourhood solidarity or social capital had an impact on patients' use of MyChart. However, the authors describe relatively little of the neighbourhood effect in their discussion. I would suggest that the authors add to the discussion other elements of the neighbourhood that may have had a positive or negative impact on the use of MyChart.

Response:  We agree with the reviewer that neighborhood solidarity or social capital is defined as “the resources accessed by individuals as a result of their membership to a network or a group.”  Social capital is a double-edged sword with the potential to promote health or harm health (Kawachi I, Berkman LF. Social capital, social cohesion and health, in Social Epidemiology; website: https://doi.org/10.1093/med/9780195377903.003.0008) and full text is available at:  (https://faculty.washington.edu/matsueda/courses/590/Readings/Kawachi%20and%20Berkman.pdf ).  However, we did not have information on social capital at the neighborhood levels.  Therefore, we have included the following text in the discussion section and address its limitations in the discussion as well. (see line 327-340)

Neighborhood-level factors such as social capital or social cohesion may affect health information technology use, including patient portal use.  For example, social capital may affect patient portal use through different pathways: 1) by providing access to patient portal use through robust internet coverage in the neighborhood; 2) by expanding access to health information technology by installing health kiosks in targeted locations (example: local departments of social service, and community health centers (Moiduddin A and Moore J, The underserved and health information technology: Issues and Opportunities. Full text is available at the website https://aspe.hhs.gov/sites/default/files/migrated_legacy_files//101521/report.pdf; 3) dissemination of resources (example: communities that are cohesive in which neighbors trust each other may diffuse the benefits of using patient portals much more easily) and 4) by providing support (neighbors may take it as a civic responsibility and provide support services for patient portal use).   Therefore, an important agenda for future research is to identify the associations of neighborhood factors such as social capital that are not readily available in secondary data sources with health information technology in general and specifically patient portal use.

Round 2

Reviewer 2 Report

I am satisfied with the author's response.

Author Response

Thanks for reviewer's positive feedback.